# Massive carbon storage in convergent margins initiated by subduction of limestone

Chunfei Chen [1,2✉], Michael W. Förster [1,3], Stephen F. Foley [1,3] & Yongsheng Liu [2✉]

Remobilization of sedimentary carbonate in subduction zones modulates arc volcanism emissions and thus Earth's climate over geological timescales. Although limestones (or chalk) are thought to be the major carbon reservoir subducted to subarc depths, their fate is still unclear. Here we present high-pressure reaction experiments between impure limestone (7.4 wt.% clay) and dunite at 1.3–2.7 GPa to constrain the melting behaviour of subducted natural limestone in contact with peridotite. The results show that although clay impurities significantly depress the solidus of limestone, melting will not occur whilst limestones are still part of the subducting slab. Buoyancy calculations suggest that most of these limestones would form solid-state diapirs intruding into the mantle wedge, resulting in limited carbon flux to the deep mantle (< ~10 Mt C y$^{-1}$). Less than 20% melting within the mantle wedge indicates that most limestones remain stable and are stored in subarc lithosphere, resulting in massive carbon storage in convergent margins considering their high carbon flux (~21.4 Mt C y$^{-1}$). Assimilation and outgassing of these carbonates during arc magma ascent may dominate the carbon flux in volcanic arcs.

[1] Department of Earth and Environmental Sciences and ARC Centre of Excellence for Core to Crust Fluid Systems, Macquarie University, North Ryde, NSW, Australia. [2] State Key Laboratory of Geological Processes and Mineral Resources, School of Earth Sciences, China University of Geosciences, Wuhan, China. [3] Research School of Earth Sciences, Australia National University, Canberra ACT, Australia. ✉email: chfchen2016@hotmail.com; yshliu@hotmail.com

Tectonic activity has been invoked as the driver of icehouse-greenhouse intervals because variations of volcanic emissions in convergent margins exert considerable control on atmospheric $CO_2$ levels and thus climate on timescales of millions of years[1,2]. Subduction has the ability to transport large amounts of carbon (modern carbon flux: 68–96 Mt C y$^{-1}$) to replenish the mantle carbon reservoir at convergent margins[3,4]. However, most of the subducted carbon has been suggested to be released beneath forearcs and arcs and relatively little carbon may be recycled to the convecting mantle[5–7]. Furthermore, substantial quantities of the released carbon were interpreted to be stored in the crust and mantle lithosphere as the amount of $CO_2$ emitted from arc volcanoes appears to be less than that subducted[5]. Understanding the behaviour of carbon in subduction zones and its storage in convergent margins is critical to grasping how tectonic activity controls the Earth's climate.

Carbon is concentrated in sediments (carbonated siliciclastic and limestone; ~60 Mt C y$^{-1}$), altered oceanic crust (~18 Mt C y$^{-1}$), and suboceanic mantle layers (1.3–10 Mt C y$^{-1}$) in modern subducting slabs (Fig. 1a)[8]. Among diverse subducted materials, the marble derived from limestone and chalk carries about 25% of this subducted carbon (~21.4 Mt C y$^{-1}$)[3,6]. Recently, Stewart and Ague[6] indicated that most of the carbon in the carbonated siliciclastic and altered oceanic crust (>90%) is released via metamorphic decarbonation reactions during infiltration of a water-bearing fluid at forearc depths (Fig. 1a). Limestone that survives this metamorphic decarbonation is an important or major carrier for transporting carbon to subarc depths (Fig. 1a)[6]. The carbonate dissolution and hydrous melting of carbonate-bearing crustal materials were advocated as two potential pathways for transferring subducted carbon to the mantle wedge at subarc depths[9–11]. However, the efficiency of these two mechanisms depends on the production of fluid and the mode of fluid percolation (channelled vs. pervasive) in dynamic subduction processes[12]. Fully hydrated oceanic crust loses about two-thirds of its initial water content in the pressure interval below 2 GPa; the efficiency of carbonate dissolution has not been investigated at higher pressures[9,13]. The fate of marble is poorly constrained. Experimental studies have largely under-evaluated the role of pure and nearly pure limestone or chalk, emphasising instead a global weighted average composition of subducted sediments (Fig. 1b), corresponding to clastic sediments with 7 wt.% carbonates[14]. This models the global sedimentary carbon contribution to subduction[15–18], but ignores the behaviour of carbonate-rich rocks that make up a substantial proportion of the subducted carbonate[3]. A number of studies have speculated that

diapirism of buoyant sedimentary materials could provide an additional pathway for recycled crustal carbonate to rise into the mantle wedge due to their low density and high melting temperature[5,15–18] but as yet, no experimental studies have investigated the fate of limestone diapirs within the mantle wedge.

In this study, we have conducted reaction experiments at 1.3–2.7 GPa and 900–1200 °C between a limestone from Ocean Discovery Program site ODP 115–714 A and a geochemically depleted peridotite (dunite) from Hannuoba, north China (see "Methods"), which were loaded as distinct, juxtaposed blocks into single experimental capsules. The limestone contains 7.4 wt.% clay and has a chemical composition similar to average global limestone and chalk (Fig. 1b). Importantly, its $H_2O$ content is similar to those of subducted marbles (Fig. 1c). These reaction experiments can be used to constrain the melting behaviour of limestone in contact with peridotite at the slab-mantle interface and of limestone diapirs during their rise through the peridotite mantle wedge. Combined with buoyancy calculations, the present study suggests that most of these limestones will not melt in subduction zones or in the mantle wedge but will form limestone diapirs that are stored in arc lithosphere in the solid-state, resulting in massive carbon storage in convergent margins.

## Results

**Phase assemblages and solidus of limestone–peridotite interaction.** The subsolidus mineral assemblages at all pressures consisted of olivine (47–48 wt.%), calcite (46–48 wt.%), and clinopyroxene (about 5 wt.%) without hydrous minerals (Supplementary Table 1, phase modes are estimated by mass balance calculation (see "Methods")). The clinopyroxenes were distributed in the former limestone layer and in the reaction zone between calcite and olivine (Fig. 2a).

Close to the solidus, the carbonatite melt (dendritic quench structure) is restricted to the contact zone between limestone and olivine (Fig. 2 and Supplementary Fig. 1). The temperature at which carbonatite melt first appears increases from 950 to 1050 °C as pressure increases from 1.3 to 2.7 GPa (Fig. 3). The proportion of carbonatite melt is 5–10 wt.% in near-solidus experiments and increases with the temperature (Supplementary Table 1; Supplementary Fig. 2). The limestone layer is entirely molten at 1200 °C and 2.0 GPa (Supplementary Fig. 1f). Interestingly, silicate melt is observed along boundaries between calcite crystals in the limestone layer away from the contact with dunite in the experiment at 2.7 GPa and 1100 °C (Supplementary Fig. 3). These silicate melts

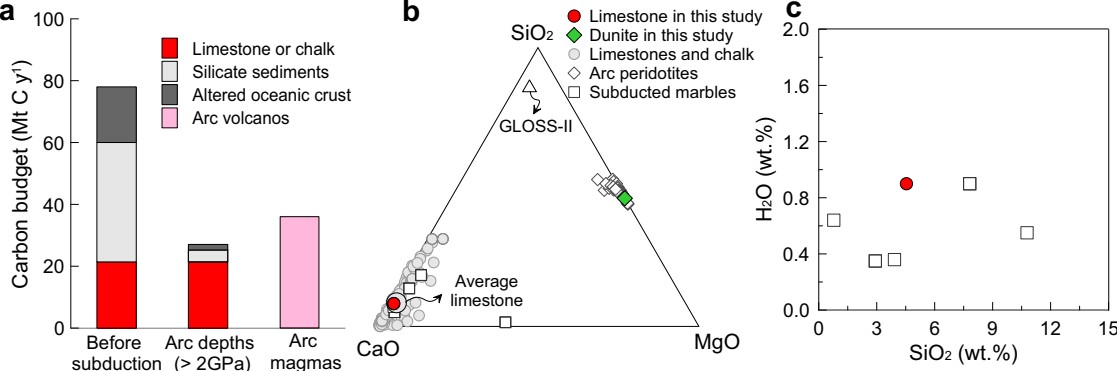

**Fig. 1 Subduction carbon fluxes and chemical compositions of starting materials. a** The constructed carbon budget in global subduction zones at forearc depths[3,8] and subarc depths[6]. The carbon emission flux in arc volcanos is shown for comparison[4]. **b** The compositions of the limestone and dunite used as the starting materials in this study compared to natural marine limestones and arc peridotites (including harzburgites and dunites). The global weighted average composition of subducted sediments (GLOSS-II) is shown for comporison[65]. Data sources of natural marine limestones and arc peridotites are provided in the supplementary information. **c** The comparison of the water content of the limestone used as the starting material with marbles from the S. W. Tianshan UHP subduction zone (>2.1 GPa)[32].

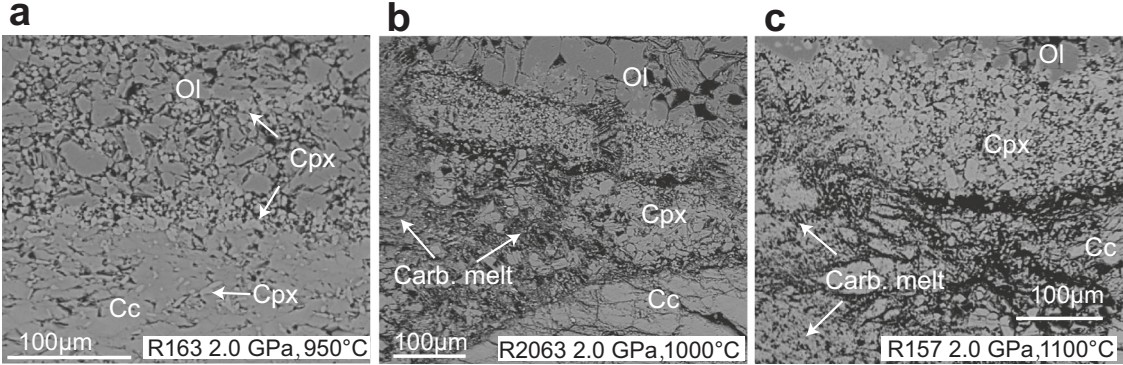

**Fig. 2 Representative backscattered electron images of experimental charges. a** Subsolidus and **b**, **c** above-solidus temperatures at 2.0 GPa. Ol olivine, Cc calcite, Cpx clinopyroxene, Carb.melt carbonatite melt.

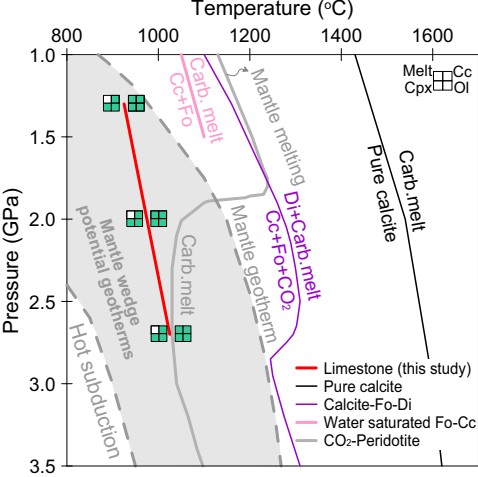

**Fig. 3 The solidus curve of limestone in contact with dunite.** The limestone solidus curve (red line: phases are shown in squares, Ol olivine, Cpx clinopyroxene, Cc calcite, and melt-carbonatite melt [top-left square]). Melting curves for pure calcite (black line[20]), the calcite–olivine–clinopyroxene system (purple line[21,22]), water-rich calcite–olivine system (pink line[23]), and peridotite + $CO_2$ (grey line[29]), are shown for comparison. The solidus of limestone in contact with dunite is about 250 °C lower than that of the calcite–olivine–clinopyroxene system and lies between the subduction zone[43] and mantle geotherms (represented by the SE Australia geotherm[66]). The grey region represents the potential geotherms of the mantle wedge, which is between hot subduction slab surface geotherms and normal continental lithosphere geotherms. Fo forsterite, Di diopside, Cc calcite.

are characterised by shoshonitic compositions with high $Na_2O$ and $K_2O$ contents (Supplementary Table 2; Supplementary Fig. 4). In contrast to the subsolidus mineral assemblages, the near-solidus experiments show a thick layer of clinopyroxene (about 150 μm) in a reaction zone between the former dunite and limestone layers (Fig. 2b). This layer formed as a reaction product as demonstrated by the occurrence of olivine inclusions in the newly grown clinopyroxenes (Supplementary Fig. 1e). The thickness of the clinopyroxene layer is almost constant with the increasing melting degrees at < 1200 °C (Fig. 2b, c) and decreases when the limestone layer is entirely molten at 1200 °C and 2 GPa. With increasing temperature, the olivine and calcite proportions decrease at all pressures.

**Phase compositions**. The clinopyroxenes in the reaction zones of the subsolidus experiments have 20.3–22.7 wt.% CaO, 14.5–18.3 wt.%

MgO, and 0.45–1.23 wt.% $Na_2O$ (Supplementary Data 1), whereas clinopyroxenes in reaction layers in the above-solidus experiments have much lower $Na_2O$ contents (0.04–0.38 wt.%). Furthermore, the compositions of clinopyroxenes in the limestone layers are significantly different from those in the reaction zones, with a gradual change from high $Al_2O_3$ content and low $CaO/Al_2O_3$ ratio in the former limestone to low $Al_2O_3$ content and high $CaO/Al_2O_3$ ratio in the clinopyroxene layer (Supplementary Fig. 4). A similar gradual change in clinopyroxene compositions occurs during the reaction of clastic sediments and peridotite[19]. Olivines show a large variation in CaO contents (mostly 0.03–1.37 wt.%), higher than those of starting materials (0.03 wt.%). Their FeO contents decrease with increasing temperature due to Fe loss to the capsule (Supplementary Table 1). Calcites in the reaction zones of the subsolidus experiments show much higher MgO contents than those within the limestone layers (e.g. 10.4–10.5 wt.% vs. 0.08–1.0 wt.% at 2.7 GPa and 1000 °C). Importantly, MgO contents of the calcites in reaction zones show a marked increase with increasing pressure (2.8–3.5 wt.% at 1.3 GPa, 8.5–9.9 wt.% at 2.0 GPa, and 10.4–10.5 wt.% at 2.7 GPa) (Fig. 4). Calcites in the limestone layers of the above-solidus experiments also have higher MgO contents than the starting materials, but lower than in calcites within the reaction zones of subsolidus experiments.

The silicate melts in the experiment at 2.7 GPa and 1100 °C have high $SiO_2$ (52.0–56.6 wt.%), $Na_2O$ (1.27–1.52 wt.%) and $K_2O$ (5.04–6.40 wt.%), and low MgO (0.5–1.5 wt.%) (Supplementary Data 1). Their major element totals are 88.2–91.2 wt.%, indicating about 10–12 wt.% volatiles (mostly $H_2O$) in the melts. The carbonatite melts rapidly quenched to glasses with no formation of silicate and carbonate minerals during quenching (Supplementary Fig. 1d). The carbonatite melts show high CaO contents (32.2–41.2 wt.%) and Ca/(Ca+Mg) ratios (atomic ratio, 0.62–0.81), and a large variation in $SiO_2$ contents (1.58–13.3 wt.%). With increasing temperature, the carbonatite melts show an increase in $SiO_2$ and MgO contents, but a decrease in CaO and $Na_2O$ contents (Fig. 4). The Ca/(Ca+Mg) ratios of near-solidus-carbonatite melts decrease from 0.81 to 0.66 with increasing pressure. Interestingly, all these melts plot on mixing lines between Mg–calcites in the reaction zones of the subsolidus experiments and olivine at various pressures (Fig. 4). Only the experimental melts at 1200 °C and 2.0 GPa plot away from the mixing lines and trend towards the clinopyroxenes.

## Discussion
Melting of pure calcite occurs at high temperatures of 1450–1620 °C at 1–3 GPa[20] and the melting loop in the model system calcite–olivine–clinopyroxene shows melting temperatures of 1100–1360 °C at 1.0–2.8 GPa[21,22] (Fig. 3). Using a hydrogen-trap technique, Weidendorfer, et al.[23] recently found that the dry solidus of the model

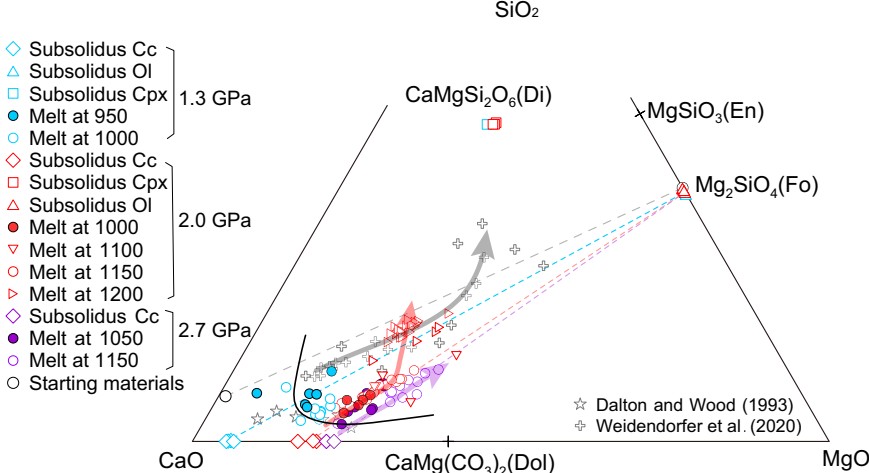

**Fig. 4 Chemical compositions of experimental products.** The compositions of carbonatite melts and minerals in the experiments. The minerals are from the reaction zones at the subsolidus experiments. The blue, red, and purple dotted lines represent mixing cures between olivine and Mg calcite in the reaction zones at various pressures. The red and purple arrows indicate the evolution of the melt compositions with increasing temperature. The grey dotted line represents the mixing cure between limestone and dunite using for this study. All these carbonatite melts plot on mixing lines between olivines and Mg calcites at various pressures, indicating no compositional contribution from clay impurities in the limestone. Carbonatite melts from water-rich calcite–olivine system[23] and carbonated peridotite system[67] are shown for comparison. Fo forsterite, En enstatite Di diopside, Dol dolomite.

system calcite–forsterite–diopside lies at 1.0 and 1.5 GPa at 1225–1250 °C and 1275–1300 °C, respectively, 140 °C higher than determined by Lee, et al.[22] and Lee, et al.[21]. They also found that the solidus of calcite–olivine–clinopyroxene with 2 wt.% water is about 1075 and 1100 °C at 1.0 and 1.5 GPa, respectively. All previous solidi determined for the model system calcite–olivine–clinopyroxene are well above the temperatures that are present within the subduction zones and mantle wedges (Fig. 3), implying that melting would occur neither in the subducting slab nor in the mantle wedge. However, a pure carbonate system is unrepresentative for natural subduction, where most carbonate sedimentary rocks contain silicate impurities, mostly clays (Fig. 1). Our study presents the first experimental results for the melting of natural sedimentary carbonate rock in contact with peridotite. Our experiments indicate that the solidus of limestone in contact with dunite lies at 900–950 °C at 1.3 GPa, 950–1000 °C at 2.0 GPa, and 1000–1050 °C at 2.7 GPa, about 250 °C lower than the solidus in the calcite–olivine–clinopyroxene system[21–23] (Fig. 3). In the following section, we discuss the melting relationships of limestone in contact with dunite and thus the applicable mechanism of low-temperature melting in the natural system.

Under subsolidus conditions, the limestone in the lower layer of the capsule experienced a metamorphic reaction: clay + calcite = Cpx (clinopyroxene) + COH-fluid. The limestone layer shows a strong chemical potential gradient to the dunite in the upper layer of the capsule, which drives chemical exchange and interaction when they are in contact with each other. The fluid released by the metamorphic reaction appears to promote the chemical interaction between olivine and calcite[23]. Mg–Ca cation exchange occurred as indicated by the formation of Mg–calcite and a thin clinopyroxene layer in the reaction zone (Fig. 2a):

$0.1$ $Mg_2SiO_4 + CaCO_3 + COH\text{-fluid}_1 = Ca_{0.9}Mg_{0.1}CO_3 + 0.1$ $CaMgSi_2O_6 + COH\text{-fluid}_2$ at 1.3 GPa;

$0.2$ $Mg_2SiO_4 + CaCO_3 + COH\text{-fluid}_1 = Ca_{0.8}Mg_{0.2}CO_3 + 0.2$ $CaMgSi_2O_6 + COH\text{-fluid}_2$ at 2.0 GPa;

$0.25$ $Mg_2SiO_4 + CaCO_3 + COH\text{-fluid}_1 = Ca_{0.75}Mg_{0.25}CO_3 +$ $0.25$ $CaMgSi_2O_6 + COH\text{-fluid}_2$ at 2.7 GPa. Interestingly, with increasing pressure, the MgO content of calcites in the reaction zone increases (Fig. 4), indicating the increasing importance of the Mg–Ca cation exchange reaction.

Compared to the subsolidus experiments, two significant features of the experiments at $T > T_{solidus}$ are that carbonatite melt occurs and the thickness of the clinopyroxene layer in the reaction zone significantly increases (the clinopyroxene mode increases from 5 wt.% to 11 wt.%) (Fig. 2b). The compositions of these solidus-carbonatite melts plot on mixing lines between Mg–calcites and olivine at their corresponding pressures (Fig. 4), indicating that they are the melting products of Mg–calcites and olivine without involvement or formation of clinopyroxene. We suggest that the reaction occurring at the solidus at various pressures is 0.1 Ol (olivine) + 0.9 Mg–calcite = carbonatite melt. As the temperature increases (T < 1200 °C), the proportion of olivine contributing to the melt increases (Fig. 4). At 1200 °C and 2.0 GPa, the clinopyroxene mode significantly decreases and carbonatite melt plots away from the mixing line between Mg–calcites and olivine and trends towards clinopyroxene (Fig. 4), indicating the dissolution of clinopyroxene in carbonatite melts. Based on the chemical compositions of the carbonatite melts and mineral modes, the melting reaction at 1200 °C and 2.0 GPa is 0.14 Ol + 0.74 Mg–calcite + 0.12 Cpx = 1.0 carbonatite melt. These observations suggest that the involvement of clinopyroxene in the melting reaction only occurs at 1200 °C. This was also observed in the model system calcite–olivine–clinopyroxene, in which melting of clinopyroxene only occurred in high-temperature experiments with high $H_2O$ content[23] (Fig. 4). The thick clinopyroxene layer in the reaction zones in the $T > T_{solidus}$ experiments could not be attributed to the melting reaction as indicated by the chemical compositions of carbonatite melts (Fig. 4) and its constant thickness with the increasing melting degrees at <1200 °C. The thick clinopyroxene reaction layer could be produced by the reaction between silicate melt derived from melting of clay, olivine, and calcite: 7.4 silicate melt + 4 Ol + 3.8 Mg–calcite = 11 Cpx + melt/fluid (enriched $H_2O$ and alkalis). This silicate melt was captured in the experiment at 1100 °C and 2.7 GPa: silicate melt is distributed along the boundaries between calcite crystals in the limestone layer that is located far from the peridotite and so represents melting of the limestone alone (Supplementary Fig. 3). This is consistent with the lower melting temperature of clay than carbonate in limestones, and also with the occurrence of initial silicate melts during melting of carbonated sediments[16,24]. These silicate melts are characterised by high CaO

and alkali contents, similar to those produced by melting of carbonated pelitic sediments (Supplementary Fig. 5) and result from the melting of clay in the former limestone. This suggests that the melting of clay in the limestone forms silicate melt that subsequently reacts with the calcite and olivine to form the thick clinopyroxene layer in the reaction zone between the limestone and dunite.

The thick clinopyroxene layer occurs together with the carbonatite melt, indicating a key role for clay in the melting of limestone in contact with dunite. Although clay impurities in limestone (7.4 wt.%) could not impart their $SiO_2$, CaO, and MgO to the carbonatite melts (no involvement of clinopyroxene in melting) (Fig. 4), the introduction of alkalis ($Na_2O$ and $K_2O$), $Al_2O_3$, and $H_2O$ to the reaction zone between the limestone and dunite through the silicate melt could significantly depress the melting temperature. The silicate melt derived from the melting of clay impurities is characterised by high alkalis ($Na_2O$ and $K_2O$) and $H_2O$ that are known to cause a profound depression of the melting point of calcite[25]. The limestone used in this study contains 0.9 wt.% $H_2O$ and the silicate melt derived from clays contains about 10 wt.% $H_2O$ (Supplementary Table 2). Although we do not know the water content in the capsule accurately due to the hydrogen diffusion through the capsule walls, the solidus between 900 and 950 °C at 1.3 GPa is much lower than in the water-saturated model system calcite–olivine–clinopyroxene from Weidendorfer, et al.[23] (Fig. 3). This could not be reasonably explained by the action of water alone and needs additional factor (s) to further depress the solidus. Phase relations in the system $Na_2CO_3$–$CaCO_3$ indicate that the addition of 10 wt.% $Na_2CO_3$–$CaCO_3$ could reduce the solidus by about 150 °C at both 0.1 GPa and 6 GPa[25,26]. The participation of silicate melts in the contact zone between the calcite and olivine will significantly reduce the minimum melting temperature through the introduction of fusible components, such as $Na_2O$, $K_2O$, $H_2O$, and $Al_2O_3$. This is indicated by high alkali contents in the near-solidus carbonatite melts ($Na_2O + K_2O = 0.58$–1.03 wt.%) (Supplementary Table 2). In summary, clay impurities play a key role in depressing the melting temperature of limestone.

Our study shows lower temperatures for the production of carbonatite melts from limestone in contact with dunite than other carbonate-bearing rocks, such as alumina-rich carbonated pelite[27] and carbonated eclogite[28] at about 3.0 GPa (the solidus lies between 1100 °C and 1200 °C). Also, no previous studies have produced carbonatite melts at <2.0 GPa in carbonated peridotite or eclogite systems (Fig. 3) because of decarbonation to produce $CO_2$ vapour at lower pressures[16,28–30]. Especially, the carbonatite melts are produced at the pressure of ≥5.0 GPa and ≥3.5 GPa in alumina-poor carbonate pelite[17] and in alumina-rich carbonated pelite[27], respectively. In contrast, our experiments indicate that limestone undergoes negligible decarbonation at low pressures, but instead can form carbonatite melt in contact with peridotite (Supplementary Fig. 1). Therefore, our data show the extension of the occurrence of carbonatite melt to shallower lithospheric mantle depths wherever limestone interacts with the mantle peridotite.

Sedimentary carbonate rocks have been suggested to experience negligible decarbonation regardless of fluid infiltration during subduction[6,31] and are the leading potential carrier of carbon to subarc depths[6]. Previous experimental investigation of the model system CaO-$Al_2O_3$-$SiO_2$-$H_2O$-$CO_2$ with 5.6–21 wt.% $H_2O$ at 4.2 and 6.0 GPa indicates that melting of hydrous pelagic limestones may occur in warm subduction zones[11]. However, natural marbles are rather dry (e.g. <1 wt.% $H_2O$ for Tianshan subducted marbles[32]). Although there are effects from fluid derived from underlying dehydrating lithologies, fluid infiltration in subducted marbles mostly occurs at low pressures of <2.0 GPa[9,33] because two-thirds of the water in the fully hydrated oceanic crust would be lost at forearc depths[12]. Furthermore,

marbles are notably impermeable to fluid flow[34,35] compared to silicate rocks, thus fluid infiltration usually occurs along a single channel in the subducted marbles[9,36]. The water content of the limestone used in our experiments is 0.9 wt.% and was probably higher during experiments due to hydrogen diffusion, thus could represent the water content of natural marbles at subarc depths. Our experimental results show that the melting temperature of limestone in contact with peridotite is higher than temperatures likely to exist at the surface of subducting slabs at the pressures studied here (1.3–2.7 GPa; Fig. 3). This implies that the melting of limestone will not occur in subducting slabs, consistent with the common view that carbonatite melts are restricted to hot regimes.

The dynamic processes of sediment subduction depend on the competing effects of their buoyancy and viscous entrainment by the subducting slab, which are controlled by the relative density and viscosity contrasts between the sediments and overlying mantle, the thickness of the sedimentary layer, and the geotherm and dynamic parameters and processes of the subducting slab (e.g. subduction rate and slab dehydration and melting)[37,38]. Abundant thermomechanical[37,39] and petrological–thermomechanical numerical calculations[38,40,41] have been developed to investigate the fate of silicate sediments using a density of 2800–3300 kg/m$^3$ and a wet quartzite rheology, all these calculations concluded that diapirism of sediments prevails at depths of about 60–120 km. We calculated the density and the viscosity of marbles and compared them to those of the overlying mantle (See "Methods" and Supplementary Figs. 6–7). Subducted marbles also are characterised by low density (2901–3028 kg/m$^3$ at <6.0 GPa) compared to the mantle peridotite (3200–3400 kg/m$^{3,42}$) and are about 100 × less viscous than wet olivine at 600–800 °C. Considering the similar low density and viscosity for both silicate sediments and marbles (Supplementary Figs. 6–7), previous numerical calculations on sediments are applicable for the marbles indicating that the subducted limestone-bearing sediment layer could detach from the downgoing slab to form buoyant diapirs.

To predict the depth of formation of a carbonate diapir during subduction, we have compiled the thickness of global subducted sediments, involving limestone or chalk (mostly > 250 m, Fig. 5a)[3], and calculated the temperatures required for the formation of a sediment diapir, involving limestone. We project these on a diagram of temperature versus sediment layer thickness from the thermomechanical calculations of Behn, et al.[37] (Fig. 5b) using a density contrast of −200 kg/m$^3$ and viscosity ratio of 1:100 between the sediment and overlying mantle (Details see "Methods"). This shows that the temperatures required for sediment diapir formation range from 600 to 805 °C (Fig. 5b), corresponding to pressures of about 2.4–3.5 GPa on slab geotherms (Fig. 5c)[43,44]. Our results predict that most sediment columns involving limestone or chalk could detach from the slab at depths of 72–105 km, consistent with previous modelling results[37–39]. Without considering the effect of dehydration and partial melting of the slab in the thermomechanical modelling, the results of our calculation probably represent the maximum depth of diapirism because petrological–thermomechanical modelling suggested that hydration of the overlying mantle would enhance carbonate diapirism[38,40,41]. Furthermore, the occurrence of clay layers within thick carbonate sequences may assist diapir formation by providing slip planes that promote detachment from the slab. Our results also indicate that the diapirism of thin sedimentary layers (<200 m) requires high temperatures, which could not be achieved during subduction. Therefore, thin marbles packaged in sedimentary columns in cold subduction zones would be subducted deep into the convecting mantle. The deep subduction of carbonates is evidenced by ultrahigh-pressure siliceous marbles as lenses in the Kokchetav Massif[45] and as thin

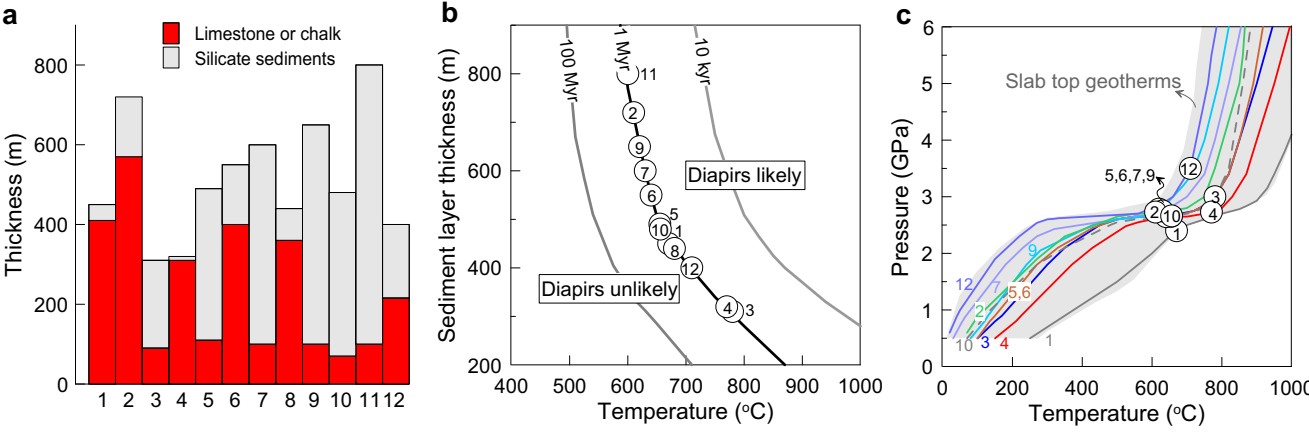

**Fig. 5 Thickness of limestone-bearing sediments and buoyancy calculations. a** The compiled thickness of subducted sediment columns involving limestone or chalk[3]. Numbers correspond to subduction zones: 1-Aegean, 2-Makran, 3-Peru, 4-Colombia-Ecuador, 5-Costa Rica, 6-Guatamala, 7-Mariana, 8-S.Luzon, 9-Solomons, 10-S. Sandwich, 11-Kamchatka, and 12-Kermadec. **b** Time needed to destabilise sediment layers expressed as temperature and sediment layer thickness for a density contrast of −200 kg m⁻³ and viscosity ratio of 1:100 between the sediment and overlying mantle from Behn, et al.[37]. The required temperatures for the initiation of diapirs from the subducted sediment columns with the limestone or chalk are obtained by projecting onto the 1 Myr line which is the time of the initiation of a sediment diapir. (**c**) Pressure–temperature paths of the slab surface for subduction scenarios with sediment columns, involving limestone or chalk[43]. We calculate the pressures for the initiation of a diapir by projecting their required temperatures onto their slab geotherms.

layers in the Sulu terrane[31] exhumed during subduction of continental crust.

The combination of numerical modelling with our experimental results constrains the behaviour and fate of limestone diapirs in the mantle wedge. Based on numerical modelling from Gerya and Meilick[38], two geodynamic regimes for the dynamics of sediments in the mantle wedge will occur (a) if the sediment diapirs do not melt, they may extend horizontally and thus underplate the lithosphere[38,39] (Fig. 6a); (b) if abundant melting occurs, they may be emplaced into subarc crust and trans-lithospheric sedimentary diapirs could weaken the continental lithosphere to generate a weak channel[38] (Fig. 6b). Which of the two geodynamic scenarios occurs also depends on the melting behaviour of other silicate sediments accompanying the limestone during subduction, as clastic sediments have lower melting temperatures than those of carbonates, as indicated by our experiments. Based on the geotherms of the mantle wedge in the two geodynamic regimes[38] (Fig. 6a, b), our experimental results predict that melting of the limestone only occurs in the scenario in Fig.6a and the degree of melting of carbonate is about 20% (Fig. 6c). The residues are mainly composed of refractory calcite, olivine, and clinopyroxene (Supplementary Table 1), which have a high solidus temperature and will therefore be stored in the mantle wedge in the solid state. In summary, we suggest that the majority of thick subducted limestones and chalk (> 80%) will be stored in arc lithosphere in the solid state.

Carbonatite metasomatism derived from recycled sedimentary carbonates is sporadically found in carbonated peridotite xenoliths in lavas from arc settings[46–48], consistent with a low degree of melting during subduction and diapirism of limestone. Importantly, our results suggest that subduction of limestone could result in the storage of abundant marble in arc lithosphere, a carbon reservoir that has been overlooked in previous studies. Excellent examples in previous publications witness that it is indeed widespread. Marbles from the Bohemian massif in the western Alps equilibrated at mantle conditions (>2.5 GPa and 1100 °C), indicating the existence of solid-state marble for >15 Ma in the mantle[49]. Carbonatite intrusions and xenoliths along the northern margin of the North China Craton (about 10 Ma), which retain many geochemical features of limestones, resulted from the subduction of limestones that formed part of the Paleo-

Asian oceanic slab at >230 Ma[50,51]. These limestone-derived carbonatites indicate that the limestone formed part of the Paleo-Asian oceanic slab that had been stored in the lithosphere mantle for at least 200 Ma. Marbles also frequently exist in the lower crust of convergent margins, such as the Calabrian massif[52] and the Bamble granulite terrane[53]. These meta-sediments in the lower crust were suggested to result from diapirism of sediments from the subduction zone[54]. These observations suggest that subduction and diapirism could transport abundant marine carbonate rocks and store them in the continental lithosphere.

Our results have important implications for the global carbon mass balance in convergent margins and subduction zones. Previous studies have suggested that about 40–65% of subducted carbon would be released via fluids or metamorphic decarbonation reactions at forearc depths (33–53 Mt C y⁻¹)[6,55]. Most of this carbon was considered to originate from carbonated sediments and altered oceanic crust[6]. Only about 35% of carbon (~28 Mt C y⁻¹) could be transported to arc magma source depths, mostly as marble (~21.4 Mt C y⁻¹)[3,6] (Figs. 1a and 6a). Our results suggest that these marbles would not be subducted into the deep mantle but instead stored in the lithosphere mantle or deep crust in the solid state. These imply that large-scale storage of carbon occurs overwhelmingly beneath continental margins or island arcs. Considering other potential carbon transfer mechanisms (such as carbonate dissolution and hydrous melting in subduction zones)[9,10], we suggest that less than 13% of carbon could be subducted to the deep mantle (< ~10 Mt C y⁻¹) (Fig. 6a).

We suggest that carbonate rocks stored in convergent margins are a key volatile source for arc volcanoes, whose emissions have been suggested to be the driver of icehouse-greenhouse intervals[1,2]. The heavier δ¹³C isotopes of global arc volcanic gas relative to the canonical mid-ocean ridge basalt value suggest that volcanic arc emissions are dominated by the remobilisation of crustal sedimentary carbonate rocks in the lithosphere during their ascent rather than carbon release by melt or fluid from subduction zone[2,56]. It is unclear how limestone could otherwise be emplaced into the deep crust or lithospheric mantle from the surface[57]. Our results indicate that the subduction and diapirism of limestone is potentially a key process for the transport and

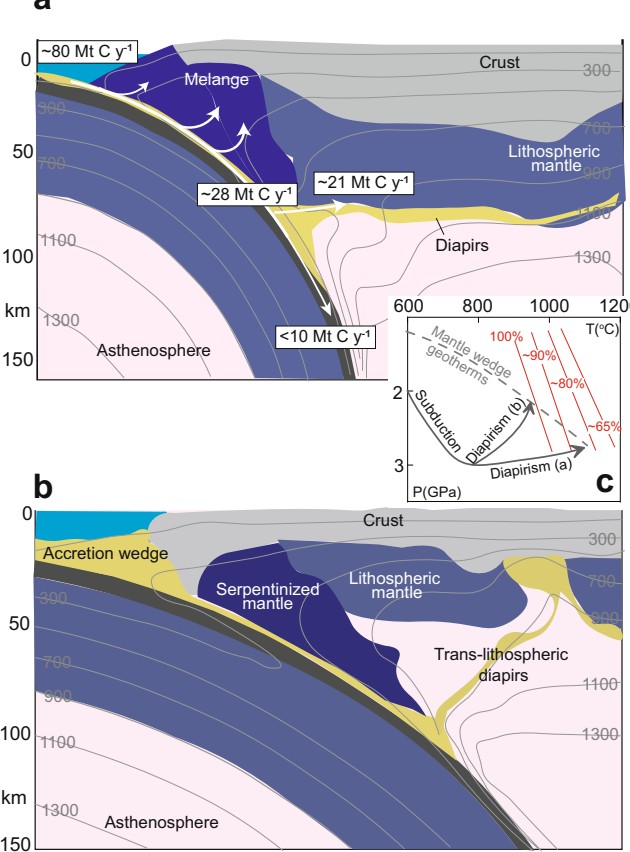

**Fig. 6 Carbon fluxes and reservoirs in convergent margins.** Schematic illustration showing two possible scenarios **a**, **b** for transfer of limestone-bearing sediments (light yellow) from the slab to the mantle wedge, superimposed on the numerical geodynamic model of Gerya and Meilick[38] at a time shot at >40 Ma from subduction initiation. In the first scenario **a** there is no extensive melting of the sediments. In the second scenario **b**, melts derived from the melting of silicate sediments rheologically weaken the lithosphere below the arc forming a weak channel into which sediment diapirs may be emplaced into subarc crust forming trans-lithospheric sedimentary plumes. The carbon fluxes (white arrows) are similar to Fig. 1a. **c** The schematic pressure–temperature (P–T) diagram of limestone that is subducted and entrained in a diapir for the two scenarios **a** and **b**. The red lines are the proportions of residual calcite during the melting of limestone in contact with the peridotites from the experimental results in this study.

storage of sedimentary carbonate in the deep crust and the lithosphere mantle in convergent margins (Fig. 6a, b). Remobilisation of these carbonate rocks during continental collisions and continental arc flare-ups could be key dynamic processes for regulating atmospheric $CO_2$ through time[1,5,57].

## Methods

**Starting materials.** The starting materials were dried at 120 °C for 48 h before loading into capsules. Their chemical compositions are listed in Supplementary Table 2. We used a natural limestone from Ocean Discovery Program Expedition 115 Site 714 A located in the northern equatorial Indian Ocean (U0714A-13H-2 W) at 5°03.6′ N and 73°47.2′ E to represent subducted limestone. This rock contains 7.4 wt.% clay and its compositions are representative of fairly pure marine limestones and subducted marbles from the S.W. Tianshan UHP subduction zone (>2.1 GPa)[32] (Fig. 1b). The analysis on the proportion and compositions of the clay in the limestone see Analytical methods. The clay in the limestone shows high $SiO_2$, $Al_2O_3$, and $Na_2O + K_2O$ contents and contains 12.1% $H_2O$ (Supplementary Table 2), similar to the chemical compositions of subducted silicate sediments[14]. The bulk $H_2O$ content of this limestone was calculated by the clay water content

and the modal abundance of clay in the limestone. The calculated water content of limestone is 0.9 wt.%, similar to those of subducted metamorphic marble from Tianshan[32] (Fig. 1c). Therefore, the limestone in this study is suitable to model the melting behaviour of carbonate rock at the slab-mantle interface and the rise of diapirs through the peridotite mantle wedge.

The dunite was made up of natural olivine (Mg# = 89.6) picked from a lherzolite xenolith from Hannuoba, China, to represent depleted peridotite in an arc setting (Fig. 1b). Although harzburgite may be more common than dunite in the mantle wedge, dunite is used here to simplify the system and so better characterise the melting reaction, involving limestone and olivine. However, the solidus of limestone in contact with dunite would be not significantly different from that in contact with harzburgite. Compared to the dunite, the harzburgite is petrographically enriched in orthopyroxene and clinopyroxene and geochemically relatively enriched in $SiO_2$, $Al_2O_3$, and CaO contents. As discussed in the main text, the melting of limestone in contact with dunite was triggered by the reaction among calcite, olivine, and the clay-derived silicate melt in the reaction zone. This reaction zone is saturated in $SiO_2$, $Al_2O_3$, and CaO, which was also confirmed by its thick clinopyroxene layer. The solidus of limestone could not be significantly modified when reacting with the harzburgite.

**High-pressure experiments.** Experiments were carried out at 1.3–2.7 GPa and 900–1200 °C using piston-cylinder apparatuses at Macquarie University (experiments at 1.3–2.0 GPa) and at the University of Mainz (experiments at 2.7 GPa). The starting materials were loaded into small capacity (2.8 mm diameter and 3–4 mm height) $Ag_{70}Pd_{30}$ or $Au_{80}Pd_{20}$ capsules (depending on temperature) for experiments at Macquarie University. Inner capsule holders and spacers were $Al_2O_3$, surrounded by a graphite heater and $CaF_2$ pressure-transmitting medium, mantled by Teflon foil. The run durations were 12–120 h (Supplementary Table 1). The capsules for experiments at the University of Mainz were 4 mm in diameter and 4–5 mm in height and made of platinum. The inner capsule holder and spacer were $Al_2O_3$ and surrounded by a graphite heater, inside a pressure-transmitting medium composed of $CaF_2$. The piston-cylinder apparatus is equipped with extra cooling channels around the tungsten carbide core within the bomb plate[58], ensuring rapid quenching of the carbonatite melts. To investigate the effect of experimental duration on melt composition, we performed three experiments of different duration (12, 24, and 48 h) at 1150 °C and 2.0 GPa.

The comparison of the compositions of carbonatite melts in the three time-series experiments (12, 24, and 48 h at 1150 °C and 2.0 GPa) indicates relatively homogeneous chemical compositions for experiments of one day or longer at 1150 °C (24 h and 48 h; Supplementary Fig. 8). This indicates that the long duration (>100 h) of most of our low-temperature experiments (<1050 °C) ensures that these near-solidus carbonatite melts are relatively homogeneous, equilibrium compositions.

**Analytical methods.** Major element analyses of olivine, clinopyroxene, and carbonatite melt were performed using a wavelength dispersive JEOL JXA 8200 microprobe at the University of Mainz. Operating conditions were 15 kV accelerating voltage and a current of 12 nA with peak counting times between 20 and 30 s for analyses of clinopyroxene, silicate melt, and carbonatite melt. Olivines were measured at 20 kV and 30 nA with peak counting times of 150 s to access minor element concentrations. The beam diameter for olivine, clinopyroxene, and silicate melt was 2 μm. Due to the chemical heterogeneity caused by quenching of carbonatite melt, we used a larger beam diameter of 10 μm for analysing major elements of carbonatite melt. A variety of minerals and synthetic materials were used as reference materials.

The major element compositions of the limestone were determined by XRF at Macquarie University. The proportion of clay was obtained by a leaching experiment using 10 wt.% acetic acid based on the weight of limestone and residual clay during the leaching experiment. The water content of the clay was based on loss on ignition of the clay at 800 °C. The major element composition of the clay was calculated by the proportion of clay and major elements of limestone through assuming that $SiO_2$, $TiO_2$, $Al_2O_3$, FeO, MnO, $Na_2O$, and $K_2O$ are in the clay and that carbonate is composed of CaO and MgO. The total content of CaO and MgO in the clay can be obtained by 100%-other major elements and water contents. To test the reliability of these calculations, we compare the calculated volatile contents of the limestone with the measured loss on ignition of the limestone (42.5 wt.%). The volatile contents of the limestone can be calculated from the $CO_2$ of the carbonate and water content of the clay. The calculated volatile content of the limestone is 41.9 wt.%, consistent with the loss on ignition of the limestone. The major element compositions of the dunite were obtained by major element compositions of olivine measured by the microprobe.

**Mass balance calculation.** The carbonate melts and calcites show similar chemical compositions with high CaO contents (Supplementary Data 1) so that it is difficult to quantify the proportion of carbonate melt in a calcite matrix by mass balance calculation. Here, the proportion of carbonatite melt is calculated based on the analysis of BSE images, whereas other mineral modes were calculated by mass balance calculation. The phase modes ($M$) were obtained by solving the following equations to minimise the $\sum R^2$:

$$C_{\text{bulk C}}^i = \sum_n^{\text{phase}}[M_n \cdot C_n^i] \tag{1}$$

$$\sum R^2 = \sum_i^{\text{component}} [C^i_{\text{bulkC}} - C^i_{\text{bulkS}}]^2 \qquad (2)$$

Where $i$ = SiO$_2$, TiO$_2$, Al$_2$O$_3$, MnO, MgO, CaO, Na$_2$O, and K$_2$O; $n$ = olivine, clinopyroxene, calcite, and melts; $C^i_{\text{bulk C}}$ = the calculated bulk concentration of component $i$ in the system; $C^i_n$ = the measured component $i$ concentration of phase $n$; $C^i_{\text{bulk S}}$ = the $i$ concentration of the starting material in the capsule. To reduce the effect of heterogeneous compositions on clinopyroxene and calcite, we obtained the average chemical composition of many analyses. In addition, calcites in the reaction zone show much higher MgO than those in the limestone layer under subsolidus conditions, so that we calculate the proportions of calcite and Mg–calcite, respectively. The detailed results are presented in Supplementary Table 1. Iron loss to Ag$_{70}$Pd$_{30}$ and Au$_{80}$Pd$_{20}$ capsules was checked through mass balance calculation: iron loss is significant (>20 wt.%) only in high-temperature runs (>1150 °C) but is limited in most subsolidus experiments and near-solidus experiments (<15 wt.%), indicating a limited effect on the solidus by iron loss.

**Details of the instability calculations.** We calculated the densities of the limestone used for the starting material in our high-pressure experiments as a function of temperature and pressure using Perple_X 6.9.0[59] and compared them to the density of the harzburgite[42]. For hot (Colombia-Ecuador) and cold (Kermadec) slab-top geotherms[43], the density contrast ranges from −348 to −228 kg/m$^3$ at <6 GPa (Supplementary Fig. 6). Interestingly, the magnitude of the density contrast significantly increases at a pressure of about 2.5 GPa, due to significantly increasing temperature along the subduction geotherm at this pressure. This indicates that limestone-bearing sediments are more likely to form the buoyant diapirs at >2.5 GPa than the silicate sediments, strengthening our calculation results of carbonate diapirs forming at the pressure of 2.4–3.5 GPa.

We calculated viscosities for dry and wet olivine[60], wet quartz[61], and calcite[62,63] over a range of temperature for a strain rate of $10^{-14}$ s$^{-1}$. Calcite is 100 × less viscous than wet olivine and slightly less viscous than wet quartz at 600–800 °C (the approximate temperature range over which diapirs form) (Supplementary Fig. 7). Considering there is no significant viscosity difference between calcite and wet quartz, the viscosity of limestone would not significantly change with the variation of its clay content.

Using thermal-mechanical modelling, Behn, et al.[37] calculated the relationship of instability time versus temperature and sediment layer thickness during subduction of sediments assuming a density contrast of −200 kg/m$^3$ and viscosity ratio of 1:100 between the sediment and overlying mantle. Their results suggested that the sediment diapirs would form when the instability time is ≤1 Ma for a subduction rate of 1–5 cm/yr. Given the density contrast of −348 to −228 kg/m$^3$ and viscosity ratio of about 1:100 between the marble and overlying mantle, we can calculate the temperature required for the formation of a carbonate diapir, projecting the thickness of limestone-bearing sediments on the 1 Ma curve in Fig. 5b. Based on the slab geotherms and the required temperatures, we can calculate the depth of formation of carbonate diapirs. The calculations are sensitive to the slab-top geotherm and diapirism may occur at a shallower depth if the subduction is hotter than the thermal model used in our calculation[64].

## Data availability
The authors declare that all data supporting the findings of this study are included within the paper and its supplementary information files.

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

## Acknowledgements

We thank Dr. John Adam, Dr. Stephan Buhre, and Dr. Slava Shcheka for assistance with experimental apparatuses, and Dr. Mingdi Gao and Dr. Xiong Wang for discussion on density and viscosity calculations. SFF and CC are funded by ARC grant FL180100134. This research is co-supported by NSFC (41530211), Key R&D Program of China (2019YFA0708400), and SKL-GPMR (MSFGPMR01). MWF is funded by Macquarie University grant MQRF0001074-2020. The Ocean Discovery Project provided the marine limestone sample.

## Author contributions

S.F.F and CC designed the study. CC and M.W.F. carried out the experiments and performed analytical measurements. CC wrote the manuscript and all authors contributed to interpreting data and revising the manuscript.

## Competing interests

The authors declare no competing interests.
