## [Peer Review File · Nature Communications]

REVIEWER COMMENTS

Reviewer #1 (Remarks to the Author):

Title: Massive carbon storage in convergent margins initiated by subduction of limestone

Authors: Chunfei Chen, Michael W. Förster, Stephen F. Foley, and Yongsheng Liu

Overview and General Recommendation:

This manuscript reports the results of high-pressure reaction experiments involving impure limestone and dunite. The authors connect these results to questions in global carbon cycling, drawing some interesting conclusions. Namely, they predict that impure limestone will not undergo melting within a subducting slab, and would experience only limited melting in the overlying mantle wedge. Thus, buoyant carbonate diapirs could be stored in the solid state in subarc lithosphere.

These results are both novel (impure limestone + peridotite has not previously been considered) and have broad, exciting implications. The paper is also very well written and clearly presented.

There are just a few limitations/clarifications that should be addressed prior to acceptance.

I recommend publication of this manuscript, provided the authors respond to the following comments.

General Comments:

The three issues noted below have in part (2/4) already been mentioned by the authors, but I believe they are substantial enough to warrant more discussion.

1) The authors note (lines 229-230) that their results predict no melting "unless abundant water is also involved." This is a bit too vague. In particular, Weidendorfer et al. (2020) find a continuous depression of the (calcite+olivine+cpx) solidus: as wt% H₂O increases, with even ~1 wt% H₂O (hardly "abundant water") having a large effect.

I understand an experiment constraining water is beyond the scope of this study, but the authors should discuss its possible influence more thoroughly. A 200 °C depression of their solidus would make melting at the slab surface a possibility. A 100 °C depression would allow melting along diapir path b (Fig. 6).

Would the authors argue that their lower-water system is a better representation of physical reality?

Or do they suspect their system (open to H infiltration) may already have a substantial water content?

Or would they expect a less profound effect of water in this clay-bearing system?

The above should be further explored.

2) Similarly, the decision to use a starting material of dunite rather than harzburgite (a better representation of mantle wedge material) is briefly mentioned in lines 447-450. Would a less-depleted starting material result in substantially different carbonate melting behavior? Perhaps the answer is not well known, but the authors should address the expected magnitude and direction of the effect (if any) this may have.

3) For the buoyancy calculations, the authors project given sediment thicknesses onto a plot from Behn et al.'s Figure 4 (2011). However, the parameters used in Behn et al.'s calculations are not necessarily appropriate to apply to marble. Specifically, it is not at all clear that a wet quartz viscosity is relevant here. Since Behn et al. report that their results are sensitive to viscosity contrast, this limitation should also be discussed. Alternatively, if the authors have evidence that the viscosities will be similar it should be presented.

4) There is an issue with the melt compositions at 1.3 GPa. The authors have drawn an up-T-trend arrow on Figure 4, but it does not seem to relate to the actual data that are plotted. If anything, they show either no significant up-T trend or perhaps an increase in MgO, falling off the dashed mixing line. In fact, the averages and standard deviations in Table S2 look indistinguishable. The lines drawn in Fig. 4 and the interpretation of the 1.3 GPa melt trends should be reassessed. This doesn't seem to have any effect on the broad conclusions, but should nevertheless be addressed.

Specific Comments:

Line 67-68: A small point, but the P-T conditions explored here do not "simulate environments... at the slab mantle interface." This is fine, since you demonstrate that no melting would occur at those lower T's anyway. But on my first reading it seemed you were implying that 900-1250 °C at 1.3-2.7 GPa could be

achieved on the slab surface. Suggest rephrasing.

Line 72-73: suggest "... during arc magma ascent is could be an important source" since this question is still debated.

Line 241-243: It is worth noting that Penniston-Dorland et al. (2015) find systematically hotter subduction geotherms from field-based evidence. If you take these higher T's as representative, diapirism would be expected at even shallower depths.

Line 251-266: This paragraph almost makes it sound like the authors conducted their own numerical model. It's spelled out very clearly in the Fig. 6 caption, but it would be helpful to mention here that they are using the PT paths from Gerya & Meilick's (2011) numerical modeling.

Line 296-306: Part of the argument around the significance of continental arcs, specifically, is that CO₂ could be assimilated/degassed from thick carbonate sections in the shallower crust of the upper plate. It is definitely valid and interesting to suggest an alternate source, but the authors should make the existence of this alternate hypothesis clear.

Line 485-488: I don't understand how the authors arrive at the clay composition. Table S2 shows 2.71 wt% between the Ca and Mg columns. Does this mean wt% Ca + wt% Mg = 2.71 ? If so, how is that determined: is that the missing wt% to make the clay total 100.0%? or is that based on Ca and Mg that can't be stoichiometrically balanced in carbonate by the inferred CO₂ of the bulk rock? Would the two methods yield internally consistent results? Please clarify

Line 490-501: can you provide an example or generalized equation of how the mass balance calculations were done?

Line 507: Should the source for the natural limestone and peridotite compositions be cited in the main text?

Line 527-534: Can you mention in the caption what the colorful arrows represent? Also, check for several typos in this caption.

Figure 2b,c: can you label carbonate melt more precisely, e.g., with an arrow or outline?

Figure 3: The gray melting curve is the only one not included in the legend. Also, the lines in the legend are too thin to really see the different colors.

Figure 4: It took me a while to figure out all the different symbols in this plot. It's purely a preference, but I think grouping the symbols by color in the legend (e.g, all the 2.7 GPa symbols grouped together) would be more helpful.

Figure 6: There is an issue with how this figure is presented, with panel c covering up important features of panels a and b. "Diapirs" is misspelled in panel a.

Table S3-1: the column labeled "LOI" appears to be mislabeled, actually the measured wt% total excluding LOI.

Figure S1: what are the lighter areas in the olivine layer of panel e?

Figure S2: Should note somewhere that the T's on the x-axis are not evenly spaced.

Reviewer #2 (Remarks to the Author):

This is a very interesting paper describing a high pressure experimental simulation of possible reactions between subducted clay-bearing limestone and overlying peridotite wedge (dunite in the experiments). The main conclusion is that melting temperatures of such an assemblage are probably too high relative to subduction geotherms and so melting will not usually occur. However, density and viscosity modelling suggests solid limestone (marble) diapirs could form and ascend from the slab surface into the peridotite wedge, leading to very significant sequestration of carbon in the wedge.

I would like to see this paper published in Nature Communications because it is novel and potentially important in relating the deep carbon cycle of the earth to tectonic processes. It will therefore be of interest to a broad community of researchers studying the deep carbon cycle, arc magmatism and so on. The experiments were well done and the data and modelling seem excellent quality. I believe it should be published, with a few relatively minor changes, as detailed below.

1. Line 34: You could and probably should cite the paper by Sieber et al. (2018) EPSL here, as it dealt with

the likely fate of CO₂+H₂O fluids released from slabs into the fore-arc region.

2. Line 108: I note that quite a bit of Fe has been lost to the AuPd, AgPd or Pt capsules during the experiments. Although the bulk FeO contents are modest, does the reduction of Fe²⁺ in the dunite to Fe (alloying with the capsule) significantly alter the H₂O content of the mix? FeO + H₂ = Fe + H₂O? If so, does this affect the solidus temperatures?

3. Line 119: The meaning of the sentence "The carbonatite melts quenched to glasses without including quenched crystals of silicate and carbonate up to several μm in size" escapes me, I am afraid - please rephrase this more clearly.

4. Line 124: "ratio" should be "ratios".

5. One variable which is not really well addressed in this study is the effect on melting temperature of H₂O content. It appears this particular limestone contains 0.90 wt% H₂O, which is presumably stored in the clay. But what is the range of H₂O contents (i.e. clay contents) of natural, subducted limestones? Is it highly variable? What is the effect on the solidus temperature of significantly higher H₂O contents? The solidus curve for H₂O-saturated calcite + olivine is listed in the key for Figure 3, but the line is either missing or too faint to see (see later comment on figures).

6. Line 165: I don't like the statement the carbonatite melts are the eutectic melting products of Mg-calcites and olivine, without involvement or formation of clinopyroxene. The fact that nearly all of the melts plot below the calcite - olivine join in Figure 4 indicates that diopside crystallisation is required for mass balance. The melts co-exist with diopside, they contain minor but measurable Al₂O₃, and are presumably at least locally in equilibrium with diopside. So they must be eutectic melts of CaCO₃+olivine+diopside, even if the diopside contribution to the melt is tiny.

7. Lines 188-191: In the experiment that formed an interstitial, broadly shoshonitic melt on grain boundaries between calcite crystals in the limestone layer, why didn't that melt react to form clinopyroxene (or at least wollastonite) crystals through out the limestone layer?

8. Several of the figures (particular 1, 3 and 4) are too faint to see clearly. They need to be redrafted with thicker lines and bolder symbols and text. Also, in Figure 2, it would be good to have some spatial context about where in the various experimental charges these images were taken.

Reviewer #3 (Remarks to the Author):

The manuscript investigates the phase equilibria of clay-bearing limestone reacting with dunite at sub-arc conditions. The authors find that clay reduces the solidus of limestone and that the impure limestone can produce carbonatitic liquids at Ts exceeding 900-1000C, from 1.3 - 2.7 GPa. The authors also perform some calculations to show that carbonate diapirs would detach from the slab and carry the carbonate to the overlying lithosphere which could potentially be a long term reservoir for carbon.

The experimental study is detailed and well explained. However, my primary criticism is about the instability calculations. The authors have used the instability calculation from Behn et al (2011) to predict diapirism in the limestone. What viscosity has been used is not clearly mentioned, and an important factor is investigating diapirism in light of geodynamics of subduction. For example, what convergence rates, thermal profile of subduction zone would affect diapirism for a given layer thickness? How does clay impurity affect viscosity in limestone? How sensitive is diapirism to deviations in viscosity from pure limestone? Without showing the effects of dynamics on diapirism, the instability formation remains a speculation. What about the effect of other layers (AOC, serpentized lithosphere) on instability formation? However, if indeed limestone does form diapirs, that would have implications for deep carbon cycle, as pointed out the by the authors.

Replies below in blue color

REVIEWER COMMENTS

Reviewer #1 (Remarks to the Author):

Title: Massive carbon storage in convergent margins initiated by subduction of limestone

Authors: Chunfei Chen, Michael W. Förster, Stephen F. Foley, and Yongsheng Liu

Overview and General Recommendation:

This manuscript reports the results of high-pressure reaction experiments involving impure limestone and dunite. The authors connect these results to questions in global carbon cycling, drawing some interesting conclusions. Namely, they predict that impure limestone will not undergo melting within a subducting slab, and would experience only limited melting in the overlying mantle wedge. Thus, buoyant carbonate diapirs could be stored in the solid state in subarc lithosphere.

These results are both novel (impure limestone + peridotite has not previously been considered) and have broad, exciting implications. The paper is also very well written and clearly presented.

There are just a few limitations/clarifications that should be addressed prior to acceptance. I recommend publication of this manuscript, provided the authors respond to the following comments.

General Comments:

The three issues noted below have in part (2/4) already been mentioned by the authors, but I believe they are substantial enough to warrant more discussion.

1) The authors note (lines 229-230) that their results predict no melting “unless abundant water is also involved.” This is a bit too vague. In particular, Weidendorfer et al. (2020) find a continuous depression of the (calcite+olivine+cpx) solidus: as wt% H₂O increases, with even ~1 wt% H₂O (hardly “abundant water”) having a large effect.

I understand an experiment constraining water is beyond the scope of this study, but the authors should discuss its possible influence more thoroughly. A 200 °C depression of their solidus would make melting at the slab surface a possibility. A 100 °C depression would allow melting along diapir path b (Fig. 6).

Would the authors argue that their lower-water system is a better representation of physical reality?

Or do they suspect their system (open to H infiltration) may already have a substantial water content?

Or would they expect a less profound effect of water in this clay-bearing system?

The above should be further explored.

Reply: Good point. Previous experimental investigation on the model system CaO-Al₂O₃-SiO₂-H₂O-CO₂ with 5.6-21 wt.% H₂O indicates that melting of hydrous pelagic limestones will occur in warm subduction zones¹. However, natural marbles are rather dry (<1 wt.% H₂O for Tianshan subducted marbles²) and notably impermeable to fluid flow^{3,4}. The water content of the limestone used in our experiments is 0.9 wt.% and was probably higher during experiments due to hydrogen diffusion, thus representing the maximum water content of natural marbles. Our experimental results show that the melting temperature of limestone in contact with peridotite is higher than temperatures likely to exist at the surface of subducting slabs at the pressures studied here (1.3-2.7 GPa; Fig. 3). This implies that melting of limestone will not occur in subducting slabs, consistent with the common view that carbonatite melts are restricted to hot regimes.

We have revised this point in the manuscript (Lines 221-226).

2) Similarly, the decision to use a starting material of dunite rather than harzburgite (a better representation of mantle wedge material) is briefly mentioned in lines 447-450. Would a less-depleted starting material result in substantially different carbonate melting behavior? Perhaps the answer is not well known, but the authors should address the expected magnitude and direction of the effect (if any) this may have.

Reply: The solidus of limestone in contact with dunite would be not significantly different from that in contact with harzburgite. Compared to the dunite, the harzburgite is enriched in Opx and Cpx and geochemically relatively enriched in SiO₂, Al₂O₃, and CaO. As discussed in the main text, melting of limestone in contact with dunite was triggered by the reaction among calcite, olivine, and the clay-derived silicate melt in the reaction zone. This reaction zone is saturated in SiO₂, Al₂O₃, and CaO which was also confirmed by its thick clinopyroxene layer. The solidus of limestone could not be modified when reacting with the harzburgite.

We have revised part of the manuscript (Lines 495-502).

3) For the buoyancy calculations, the authors project given sediment thicknesses onto a plot from Behn et al.'s Figure 4 (2011). However, the parameters used in Behn et al.'s calculations are not necessarily appropriate to apply to marble. Specifically, it is not at all clear that a wet quartz viscosity is relevant here. Since Behn et al. report that their results are sensitive to viscosity contrast, this limitation should also be discussed. Alternatively, if the authors have evidence that the viscosities will be similar it should be presented.

Reply: We calculated viscosities for wet olivine⁵, wet quartz⁶, and calcite^{7,8} over a range of temperature at a strain rate of 10⁻¹⁴ s⁻¹. Calcite is about 100 × less viscous than wet olivine and slightly lower than wet quartz at 600-800 °C (the approximate temperature range over which diapirs form). Behn, et al.⁹ calculated the relationship of instability time versus temperature and sediment layer thickness during subduction of sediments assuming a density contrast of -200 kg/m³ and viscosity ratio of 1:100 between sediment and overlying mantle. Their results suggested that the sediment diapirs would form when the instability lasts ≤1 Ma for a subduction rate of 1 to 5 cm/yr. Given the density contrast of -348 to -228 kg/m³ and viscosity ratio of about 1:100 between marble and overlying mantle, we can calculate the temperatures required for the formation of a carbonate diapir, projecting of the thickness of limestone-bearing sediments on the 1 Ma curve on this diagram.

For details, see Lines 242 and 577-582 of the revised manuscript.

4) There is an issue with the melt compositions at 1.3 GPa. The authors have drawn an up-T-trend arrow on Figure 4, but it does not seem to relate to the actual data that are plotted. If anything, they show either no significant up-T trend or perhaps an increase in MgO, falling off the dashed mixing line. In fact, the averages and standard deviations in Table S2 look indistinguishable. The lines drawn in Fig. 4 and the interpretation of the 1.3 GPa melt trends should be reassessed. This doesn't seem to have any effect on the broad conclusions, but should nevertheless be addressed.

Reply: We have deleted the presumed trend in the original manuscript. The mixing line is ok because these melts are not far from the mixing line.

Specific Comments:

Line 67-68: A small point, but the P-T conditions explored here do not “simulate environments... at the slab mantle interface.” This is fine, since you demonstrate that no melting would occur at those lower T’s anyway. But on my first reading it seemed you were implying that 900-1250 °C at 1.3-2.7 GPa could be achieved on the slab surface. Suggest rephrasing.

Reply: We have rephrased the sentence in Lines 68-70.

Line 72-73: suggest “... during arc magma ascent is could be an important source” since this question is still debated.

Reply: To avoid ambiguity, we have deleted this sentence.

Line 241-243: It is worth noting that Penniston-Dorland et al. (2015) find systematically hotter subduction geotherms from field-based evidence. If you take these higher T’s as representative, diapirism would be expected at even shallower depths.

Reply: We have noted this point in the revised manuscript (Lines 591-593). The calculations are sensitive to the slab-top geotherm and diapirism may occur at a shallower depth if subduction is hotter than the thermal model used in our calculation¹⁰.

Line 251-266: This paragraph almost makes it sound like the authors conducted their own numerical model. It’s spelled out very clearly in the Fig. 6 caption, but it would be helpful to mention here that they are using the PT paths from Gerya & Meilick’s (2011) numerical modeling.

Reply: We have emphasized this point in Lines 268-269.

Line 296-306: Part of the argument around the significance of continental arcs, specifically, is that CO₂ could be assimilated/degassed from thick carbonate sections in the shallower crust of the upper plate. It is definitely valid and interesting to suggest an alternate source, but the authors should make the existence of this alternate hypothesis clear.

Reply: We have clarified this point in Lines 315-316.

Line 485-488: I don’t understand how the authors arrive at the clay composition. Table S2 shows 2.71 wt% between the Ca and Mg columns. Does this mean wt% Ca + wt% Mg = 2.71 ? If so, how is that determined: is that the missing wt% to make the clay total 100.0%? or is that based on Ca and Mg that can’t be stoichiometrically balanced in carbonate by the inferred CO₂ of the bulk rock? Would the two methods yield internally consistent results? Please clarify

Reply: The total content of CaO and MgO in the clay can be obtained by 100%-other major elements and water contents. To test the reliability of these calculations, we compare the calculated volatile contents of the limestone with the measured loss on ignition for the limestone (42.5%). The volatile contents of the limestone can be calculated using the CO₂ of the carbonate and water content of the clay. The calculated volatile content of the limestone is 41.9 wt.%, consistent with the loss on ignition of the limestone. We have added this at Lines 538-543.

Line 490-501: can you provide an example or generalized equation of how the mass balance calculations were done?

Reply: We have provided the equations for mass balance calculations (Lines 551-558).

Line 507: Should the source for the natural limestone and peridotite compositions be cited in

the main text?

Reply: We have cited the detailed information in the introduction (Line 64).

Line 527-534: Can you mention in the caption what the colorful arrows represent? Also, check for several typos in this caption.

Reply: Done.

Figure 2b,c: can you label carbonate melt more precisely, e.g., with an arrow or outline?

Reply: Done.

Figure 3: The gray melting curve is the only one not included in the legend. Also, the lines in the legend are too thin to really see the different colors.

Reply: Done.

Figure 4: It took me a while to figure out all the different symbols in this plot. It's purely a preference, but I think grouping the symbols by color in the legend (e.g, all the 2.7 GPa symbols grouped together) would be more helpful.

Reply: Done.

Figure 6: There is an issue with how this figure is presented, with panel c covering up important features of panels a and b. "Diapirs" is misspelled in panel a.

Reply: We have modified the positions of these figures.

Table S3-1: the column labeled "LOI" appears to be mislabeled, actually the measured wt% total excluding LOI.

Reply: We have revised it.

Figure S1: what are the lighter areas in the olivine layer of panel e?

Reply: It is Cpx and we have marked it in the figure.

Figure S2: Should note somewhere that the T's on the x-axis are not evenly spaced.

Reply: We have noted this point in the figure caption.

Reviewer #2 (Remarks to the Author):

This is a very interesting paper describing a high pressure experimental simulation of possible reactions between subducted clay-bearing limestone and overlying peridotite wedge (dunite in the experiments). The main conclusion is that melting temperatures of such an assemblage are probably too high relative to subduction geotherms and so melting will not usually occur. However, density and viscosity modelling suggests solid limestone (marble) diapirs could form and ascend from the slab surface into the peridotite wedge, leading to very significant sequestration of carbon in the wedge.

I would like to see this paper published in Nature Communications because it is novel and potentially important in relating the deep carbon cycle of the earth to tectonic processes. It will therefore be of interest to a broad community of researchers studying the deep carbon cycle, arc magmatism and so on. The experiments were well done and the data and modelling seem excellent quality. I believe it should be published, with a few relatively minor changes, as detailed below.

1. Line 34: You could and probably should cite the paper by Sieber et al. (2018) EPSL here, as it dealt with the likely fate of CO₂+H₂O fluids released from slabs into the fore-arc region.
Reply: Done (Line 35).

2. Line 108: I note that quite a bit of Fe has been lost to the AuPd, AgPd or Pt capsules during the experiments. Although the bulk FeO contents are modest, does the reduction of Fe²⁺ in the dunite to Fe (alloying with the capsule) significantly alter the H₂O content of the mix? FeO + H₂ = Fe + H₂O? If so, does this affect the solidus temperatures?

Reply: Good point. Indeed, we do not know the water content in the capsule accurately due to hydrogen diffusion through the capsule walls. However, we suggest that this would not affect the solidus, because significant Fe loss did not occur in most of the subsolidus experiments and near-solidus experiments (<15%). We explain this point in the revised manuscript (Lines 564-566).

Furthermore, we have emphasized that natural marbles are rather dry (e.g., <1 wt.% H₂O for Tianshan subducted marbles²) and notably impermeable to fluid flow^{3,4}. The water content of the limestone used in our experiments is 0.9 wt.% and was probably higher during experiments due to hydrogen diffusion, thus representing the maximum water content of natural marbles (Lines 221-226 in the revised manuscript).

3. Line 119: The meaning of the sentence "The carbonatite melts quenched to glasses without including quenched crystals of silicate and carbonate up to several μm in size" escapes me, I am afraid - please rephrase this more clearly.

Reply: We have rephrased this sentence. The carbonatite melts rapidly quenched to glasses with no formation of silicate and carbonate minerals during quenching (Lines 118-119).

4. Line 124: "ratio" should be "ratios".
Reply: Done.

5. One variable which is not really well addressed in this study is the effect on melting temperature of H₂O content. It appears this particular limestone contains 0.90 wt% H₂O, which is presumably stored in the clay. But what is the range of H₂O contents (i.e. clay contents) of natural, subducted limestones? Is it highly variable? What is the effect on the solidus temperature of significantly higher H₂O contents? The solidus curve for H₂O-saturated calcite + olivine is listed in the key for Figure 3, but the line is either missing or too faint to see (see later comment on figures).

Reply: Firstly, the water content of the natural limestone has not been reported. Natural limestones would lose most of their water in the pressure interval below 2 GPa during dehydration of fully hydrated oceanic crust. Natural marbles (i.e. metamorphic limestones) are rather dry (e.g., <1 wt.% H₂O for Tianshan subducted marbles²) and notably impermeable to fluid flow^{3,4}. The water content of the limestone used in our experiments is 0.9 wt.% and was probably higher during experiments due to hydrogen diffusion, thus representing the maximum water content of natural marbles. Our experimental results show that the melting temperature of limestone in contact with peridotite is higher than temperatures likely to exist at the surface of subducting slabs at the pressures studied here. We have clarified this point in the revised manuscript (Lines 221-226). In addition, we have revised Figure 3 to make it clearer.

6. Line 165: I don't like the statement the carbonatite melts are the eutectic melting products

of Mg-calcites and olivine, without involvement or formation of clinopyroxene. The fact that nearly all of the melts plot below the calcite - olivine join in Figure 4 indicates that diopside crystallisation is required for mass balance. The melts co-exist with diopside, they contain minor but measurable Al_2O_3 , and are presumably at least locally in equilibrium with diopside. So they must be eutectic melts of $CaCO_3$ +olivine+diopside, even if the diopside contribution to the melt is tiny.

Reply: We have deleted the word “eutectic”.

7. Lines 188-191: In the experiment that formed an interstitial, broadly shoshonitic melt on grain boundaries between calcite crystals in the limestone layer, why didn't that melt react to form clinopyroxene (or at least wollastonite) crystals through out the limestone layer?

Reply: The formation of wollastonite requires extremely high temperature ($> 1500\text{ }^\circ\text{C}$ at 2.7 GPa)¹¹. The SEM image in Supplementary Fig. 3b indeed suggests no Cpx (or wollastonite) formation in the unreacted limestone layer. The silicate melt is depleted Mg (Table S2). No reaction of silicate melts and calcite in the unreacted limestone layer may be attributed to too low Mg to form Cpx. This also is suggested by the occurrence of Cpx in the Mg-rich reacted limestone layer in Supplementary Fig. 3a.

8. Several of the figures (particular 1, 3 and 4) are too faint to see clearly. They need to be redrafted with thicker lines and bolder symbols and text. Also, in Figure 2, it would be good to have some spatial context about where in the various experimental charges these images were taken.

Reply: We have modified the figures to improve them.

Reviewer #3 (Remarks to the Author):

The manuscript investigates the phase equilibria of clay-bearing limestone reacting with dunite at sub-arc conditions. The authors find that clay reduces the solidus of limestone and that the impure limestone can produce carbonatitic liquids at T_s exceeding 900-1000C, from 1.3 - 2.7 GPa. The authors also perform some calculations to show that carbonate diapirs would detach from the slab and carry the carbonate to the overlying lithosphere which could potentially be a long term reservoir for carbon.

The experimental study is detailed and well explained. However, my primary criticism is about the instability calculations. The authors have used the instability calculation from Behn et al (2011) to predict diapirism in the limestone. What viscosity has been used is not clearly mentioned, and an important factor is investigating diapirism in light of geodynamics of subduction. For example, what convergence rates, thermal profile of subduction zone would affect diapirism for a given layer thickness? How does clay impurity affect viscosity in limestone? How sensitive is diapirism to deviations in viscosity from pure limestone? Without showing the effects of dynamics on diapirism, the instability formation remains a speculation. What about the effect of other layers (AOC, serpentinized lithosphere) on instability formation? However, if indeed limestone does form diapirs, that would have implications for deep carbon cycle, as pointed out the by the authors.

Reply: Thanks for your comments. We have added more details and performed more instability calculations to clarify the various effects in the revised manuscript (Lines 231-261 and 567-593).

We calculated viscosities for wet olivine⁵, wet quartz⁶, and calcite^{7,8} over a range of temperatures at a strain rate of 10^{-14} s^{-1} . Calcite is about $100 \times$ less viscous than wet olivine and slightly lower than wet quartz at 600-800 °C (the approximate temperature range over which diapirs form). **Considering that there is no significant viscosity difference between calcite and wet quartz, the viscosity of limestone would not significantly change with variation of its clay content.** We also calculated the densities of the limestone used for the starting material in our high-pressure experiment as a function of temperature and pressure using Perple_X 6.9.0¹² and compared them to the density of the harzburgite¹³. For hot (Colombia-Ecuador) and cold (Kermadec) slab-top geotherms, the density contrasts range from -348 to -228 kg/m^3 at < 6 GPa.

Abundant thermomechanical^{9,14} and petrological–thermomechanical numerical calculations¹⁵⁻¹⁷ have been developed to investigate the fate of silicate sediments using a density of 2800-3300 kg/m^3 and a wet quartzite rheology. The thermomechanical models focused on the effect of variation of temperature, sediment layer thickness, and density on dynamic processes. The coupled petrological–thermomechanical model specifies many rock properties that vary as a function of pressure and temperature assuming a fixed sediment thickness and investigates the effect of dehydration and melting of the subducting slab. All these calculations concluded that the diapirism of sediments prevails at depths of about 60–120 km. Subducted marbles also are characterized by low density (2901-3028 kg/m^3 at < 6 GPa) compared to mantle peridotite (3200-3400 kg/m^3 ¹³) and are about $100 \times$ less viscous than wet olivine at 600-800 °C (Supplementary Fig. 7). Given the density contrast of -348 to -228 kg/m^3 and viscosity ratio of about 1:100 between marble and overlying mantle, all previous numerical calculations on sediments are applicable for the marbles indicating that the subducted limestone-bearing sediment layer could detach from the downgoing slab to form the buoyant diapirs.

Using the thermal-mechanical modelling, Behn, et al.⁹ calculated the relationship of instability time versus temperature and sediment layer thickness during subduction of sediments assuming a density contrast of -200 kg/m^3 and viscosity ratio of 1:100 between sediment and overlying mantle. Their results suggested that sediment diapirs would form when the instability lasts ≤ 1 Ma for **a subduction rate of 1 to 5 cm/yr**. Given the density contrast of -348 to -228 kg/m^3 and viscosity ratio of about 1:100 between marble and overlying mantle, we can calculate the temperatures required for the formation of a carbonate diapir, projecting the thickness of limestone-bearing sediments on the 1 Ma curve on this diagram. Based on the slab geotherms and the required temperatures, we can calculate the depth of formation of a carbonate diapir. The calculations are sensitive to **the slab-top geotherm** and diapirism may occur at a shallower depth if the subductions are hotter than the thermal model used in our calculation¹⁰.

Petrological–thermomechanical modelling suggested that hydration metasomatism of the overlying mantle by dehydration and partial melting of slab (other layers: AOC and serpentinized lithosphere) could enhance carbonate diapirism¹⁵⁻¹⁷. Thus, the results of our calculations probably correspond to the maximum diapirism depth **without considering the effect of dehydration and partial melting of the slab (other layers: AOC and serpentinized lithosphere)** in the thermal-mechanical modeling.

Interestingly, the density contrasts significantly decrease at the pressure of about 2.5 GPa (Supplementary Fig. 6), due to significantly increasing temperature at this pressure. This indicates that **limestone-bearing sediments are more likely to form the buoyant diapirs at >2.5 GPa than the silicate sediments** (Supplementary Fig. 6), strengthening our results for carbonate diapirs forming at the pressure of 2.4-3.5 GPa.

References:

1. Schettino, E. & Poli, S. in *Carbon in Earth's Interior* 209-221 (2020).
2. Tao, R., Zhang, L., Li, S., Zhu, J. & Ke, S. Significant contrast in the Mg-C-O isotopes of carbonate between carbonated eclogite and marble from the S.W. Tianshan UHP subduction zone: Evidence for two sources of recycled carbon. *Chemical Geology* **483**, 65-77 (2018).
3. Galvez, M. E. & Pubellier, M. in *Deep carbon: Past to present* 276-312 (Cambridge University Press, 2019).
4. Bickle, M., Chapman, H., Wickham, S. & Peters, M. T. Strontium and oxygen isotope profiles across marble-silicate contacts, Lizzies Basin, East Humboldt Range, Nevada: constraints on metamorphic permeability contrasts and fluid flow. *Contributions to Mineralogy and Petrology* **121**, 400-413 (1995).
5. Hirth, G. & Kohlstedt, D. Rheology of the upper mantle and the mantle wedge: A view from the experimentalists. *Geophysical Monograph-American Geophysical Union* **138**, 83-106 (2003).
6. Hirth, G., Teyssier, C. & Dunlap, J. W. An evaluation of quartzite flow laws based on comparisons between experimentally and naturally deformed rocks. *International Journal of Earth Sciences* **90**, 77-87 (2001).
7. Schmid, S. M., Paterson, M. S. & Boland, J. N. High temperature flow and dynamic recrystallization in carrara marble. *Tectonophysics* **65**, 245-280 (1980).
8. Heard, H. C. & Raleigh, C. B. Steady-State Flow in Marble at 500° to 800°C. *GSA Bulletin* **83**, 935-956 (1972).
9. Behn, M. D., Kelemen, P. B., Hirth, G., Hacker, B. R. & Massonne, H.-J. Diapirs as the source of the sediment signature in arc lavas. *Nature Geoscience* **4**, 641-646 (2011).
10. Penniston-Dorland, S. C., Kohn, M. J. & Manning, C. E. The global range of subduction zone thermal structures from exhumed blueschists and eclogites: Rocks are hotter than models. *Earth and Planetary Science Letters* **428**, 243-254 (2015).
11. Lee, C.-T. A. & Lackey, J. S. Global Continental Arc Flare-ups and Their Relation to Long-Term Greenhouse Conditions. *Elements* **11**, 125-130 (2015).
12. Connolly, J. A. D. Computation of phase equilibria by linear programming: A tool for geodynamic modeling and its application to subduction zone decarbonation. *Earth and Planetary Science Letters* **236**, 524-541 (2005).
13. Jull, M. & Kelemen, P. On the conditions for lower crustal convective instability. *JOURNAL OF GEOPHYSICAL RESEARCH* **106**, 6423-6446 (2001).
14. Currie, C. A., Beaumont, C. & Huismans, R. S. The fate of subducted sediments: A case for backarc intrusion and underplating. *Geology* **35**, 1111-1114 (2007).
15. Gorczyk, W., Gerya, T. V., Connolly, J. A. D., Yuen, D. A. & Rudolph, M. Large-scale rigid-body rotation in the mantle wedge and its implications for seismic tomography. *Geochemistry, Geophysics, Geosystems* **7** (2006).
16. Gerya, T. V., Connolly, J. A., Yuen, D. A., Gorczyk, W. & Capel, A. M. Seismic

implications of mantle wedge plumes. *Physics of the Earth and Planetary Interiors* **156**, 59-74 (2006).

17. Gerya, T. V. & Meilick, F. I. Geodynamic regimes of subduction under an active margin: effects of rheological weakening by fluids and melts. *Journal of Metamorphic Geology* **29**, 7-31 (2011).

REVIEWERS' COMMENTS

Reviewer #1 (Remarks to the Author):

General Comments

I am overall happy with the changes the authors have made to this revised manuscript. In particular, the treatment of the buoyancy calculations is greatly improved.

I am not completely satisfied with the response to the question of H₂O (my general comment 1 and also noted by reviewer #2, their comment 2).

It is not widely agreed that all marbles are "impermeable to fluid flow." See, for example, evidence for pervasive fluid flow in subducted marbles from Syros (Schumacher et al. 2008) and the Western Alps (Ballèvre & Lagabrielle, 1994). There is also ample evidence of channelized fluid flow through marbles (Guo et al. 2020; Ague & Nicolescu 2014). Thus the assertion that 0.9 wt% H₂O (or whatever was actually achieved in experiments open to H) represents a maximum water content is not reasonable. I still suggest the authors add a sentence or two addressing this.

Specific Comments

Line 553-558: check for typo, "bulk" vs "bluk"

Line 572-576: Cool result! But "the density contrast significantly decreases" is confusing. The density contrast becomes more negative, right? So I would say the magnitude of the density contrast is actually increasing. Suggest rephrasing.

References

Schumacher, J. C., Brady, J. B., Cheney, J. T., & Tonnsen, R. R. (2008). Glaucofane-bearing marbles on Syros, Greece. *Journal of Petrology*, 49(9), 1667-1686.

Balleve, M., & Lagabrielle, Y. (1994). Garnet in blueschist-facies marbles from the Queyras unit (Western Alps): its occurrence and its significance. *Schweizerische Mineralogische und Petrographische Mitteilungen*, 74(2), 203-212.

Guo, S., Chu, X., Hermann, J., Chen, Y., Li, Q., Wu, F., ... & Sein, K. (2021). Multiple episodes of fluid infiltration along a single metasomatic channel in metacarbonates (Mogok metamorphic belt, Myanmar) and implications for CO₂ release in orogenic belts. *Journal of Geophysical Research: Solid Earth*, 126(1), e2020JB020988.

Ague, J. J., & Nicolescu, S. (2014). Carbon dioxide released from subduction zones by fluid-mediated reactions. *Nature Geoscience*, 7(5), 355-360.

Reviewer #2 (Remarks to the Author):

The authors have addressed the points of the reviewers very well in the revised manuscript. I would be happy to see it published in its current form. I think it's an excellent and important contribution.

Reviewer #3 (Remarks to the Author):

I appreciate that the authors calculated viscosity and density of calcite and examined the effect of clay content on the rheology. But given this is a submission for a high impact journal such as *Nature Communications*, I really think it requires demonstration of diapir formation and its fate by actual thermal

mechanical modeling rather than superimposing the rheology calculations and instability timescales on an existing simulation by Gerya and Meilick (Fig 6). This is because, to me, the hook point for submitting it to Nature Communications hinges on the diapirism aspect of subducting carbonate layers, and if that's the case, diapirism should be clearly demonstrated by geodynamic modeling

REVIEWERS' COMMENTS

Reviewer #1 (Remarks to the Author):

General Comments

I am overall happy with the changes the authors have made to this revised manuscript. In particular, the treatment of the buoyancy calculations is greatly improved.

I am not completely satisfied with the response to the question of H₂O (my general comment 1 and also noted by reviewer #2, their comment 2).

It is not widely agreed that all marbles are “impermeable to fluid flow.” See, for example, evidence for pervasive fluid flow in subducted marbles from Syros (Schumacher et al. 2008)

and the Western Alps (Ballèvre & Lagabrielle, 1994). There is also ample evidence of channelized fluid flow through marbles (Guo et al. 2020; Ague & Nicolescu 2014). Thus the assertion that 0.9 wt% H₂O (or whatever was actually achieved in experiments open to H) represents a maximum water content is not reasonable. I still suggest the authors add a sentence or two addressing this.

References

- Schumacher, J. C., Brady, J. B., Cheney, J. T., & Tonnsen, R. R. (2008). Glaucofane-bearing marbles on Syros, Greece. *Journal of Petrology*, 49(9), 1667-1686.
- Balleve, M., & Lagabrielle, Y. (1994). Garnet in blueschist-facies marbles from the Queyras unit (Western Alps): its occurrence and its significance. *Schweizerische Mineralogische und Petrographische Mitteilungen*, 74(2), 203-212.
- Guo, S., Chu, X., Hermann, J., Chen, Y., Li, Q., Wu, F., ... & Sein, K. (2021). Multiple episodes of fluid infiltration along a single metasomatic channel in metacarbonates (Mogok metamorphic belt, Myanmar) and implications for CO₂ release in orogenic belts. *Journal of Geophysical Research: Solid Earth*, 126(1), e2020JB020988.
- Ague, J. J., & Nicolescu, S. (2014). Carbon dioxide released from subduction zones by fluid-mediated reactions. *Nature Geoscience*, 7(5), 355-360.

Reply: Thanks for your comments. Although there are effects from fluid derived from underlying dehydrating lithologies, fluid infiltration in subducted marbles mostly occurs at low pressure of < 2.0 GPa^{1,2} (as suggested by the marbles mentioned in the reviewer's comments), because two-thirds of the water in the fully hydrated oceanic crust would be lost at forearc depths³. Furthermore, marbles are notably impermeable to fluid flow^{4,5} compared to silicate rocks, thus fluid infiltration usually occurs along a single channel in subducted marbles^{1,6}. The water content of the limestone used in our experiments is 0.9 wt.% and was probably higher during experiments due to hydrogen diffusion, thus could represent the water content of natural marbles at subarc depths. Note that we have changed "represents a maximum water content of natural marbles" to "could represent the water content of natural marbles at subarc depths". We have revised the text at Lines 222-229.

Specific Comments

Line 553-558: check for typo, "bulk" vs "bluk"

Reply: Revised.

Line 572-576: Cool result! But "the density contrast significantly decreases" is confusing. The density contrast becomes more negative, right? So I would say the magnitude of the density contrast is actually increasing. Suggest rephrasing.

Reply: We have rephrased it.

Reviewer #2 (Remarks to the Author):

The authors have addressed the points of the reviewers very well in the revised manuscript. I would be happy to see it published in its current form. I think it's an excellent and important contribution.

Reply: Thanks for your review.

Reviewer #3 (Remarks to the Author):

I appreciate that the authors calculated viscosity and density of calcite and examined the effect of clay content on the rheology. But given this is a submission for a high impact journal such as Nature Communications, I really think it requires demonstration of diapir formation and its fate by actual thermal mechanical modeling rather than superimposing the rheology calculations and instability timescales on an existing simulation by Gerya and Meilick (Fig 6). This is because, to me, the hook point for submitting it to Nature Communications hinges on

the diapirism aspect of subducting carbonate layers, and if that's the case, diapirism should be clearly demonstrated by geodynamic modeling.

Reply: Thanks for your comments. The calculations presented in our manuscript are sufficient to demonstrate the limestone diapir formation at the pressure of 2.4-3.5 GPa. Our density and viscosity calculations suggest that the marble has a slightly lower density and less viscosity than the sediments. All previous numerical calculations on sediments are applicable for the marbles. It is reasonable to calculate the diapir depths using previous modeling results. Furthermore, the primary focus of our manuscript is to present comprehensive experimental petrological studies and not to optimize the thermal-mechanical models of others. We feel the treatment we have given to the modelling aspect is adequate.

References

1. Ague, J. J. & Nicolescu, S. Carbon dioxide released from subduction zones by fluid-mediated reactions. *Nature Geoscience* **7**, 355-360 (2014).
2. Schumacher, J. C., Brady, J. B., Cheney, J. T. & Tonnsen, R. R. Glaucophane-bearing Marbles on Syros, Greece. *Journal of Petrology* **49**, 1667-1686 (2008).
3. Schmidt, M. W. & Poli, S. in *Treatise on Geochemistry (Second Edition)* (eds Heinrich D. Holland & Karl K. Turekian) 669-701 (Elsevier, 2014).
4. Galvez, M. E. & Pubellier, M. in *Deep carbon: Past to present* 276-312 (Cambridge University Press, 2019).
5. Bickle, M., Chapman, H., Wickham, S. & Peters, M. T. Strontium and oxygen isotope profiles across marble-silicate contacts, Lizzies Basin, East Humboldt Range, Nevada: constraints on metamorphic permeability contrasts and fluid flow. *Contributions to Mineralogy and Petrology* **121**, 400-413 (1995).
6. Guo, S. *et al.* Multiple Episodes of Fluid Infiltration Along a Single Metasomatic Channel in Metacarbonates (Mogok Metamorphic Belt, Myanmar) and Implications for CO₂ Release in Orogenic Belts. *Journal of Geophysical Research: Solid Earth* **126**, e2020JB020988 (2021).